# Efficiency Pentathlon: A Standardized Benchmark for Efficiency Evaluation

## Abstract

Rising computational demands of modern natural language processing (NLP) systems have increased the barrier to entry for cutting-edge research while posing serious environmental concerns. Yet, progress on model efficiency has been impeded by practical challenges in model evaluation and comparison. For example, hardware is challenging to control due to disparate levels of accessibility across different institutions. Moreover, improvements in metrics such as FLOPs often fail to translate to progress in real-world applications. In response, we introduce efficiency Pentathlon, a benchmark for holistic and realistic evaluation of model efficiency. Pentathlon focuses on inference, which accounts for a majority of the compute in a model's lifecycle. It offers a strictly-controlled hardware platform, and is designed to mirror real-world applications scenarios. It incorporates a suite of metrics that target different aspects of efficiency, including latency, throughput, memory overhead, number of parameters, and energy consumption, hence the name **Penta**thlon. It also comes with a software library that can be seamlessly integrated into any codebase and enable evaluation. As a standardized and centralized evaluation platform, Pentathlon can drastically reduce the workload to make fair and reproducible efficiency comparisons. While initially focused on natural language processing (NLP) models, Pentathlon is designed to allow flexible extension to other fields. We envision Pentathlon will stimulate algorithmic innovations in building efficient models, and foster an increased awareness of the social and environmental implications in the development of future-generation NLP models.

## 1 Introduction

The remarkable recent progress in artificial intelligence owes much to advances in large-scale deep learning models (Brown et al., 2020; Chowdhery et al., 2022; Thoppilan et al., 2022, *inter alia*). However, their rapidly-increasing computational demands have introduced substantial challenges. The barrier to entry to cutting-edge research is raised, particularly impacting researchers and practitioners with fewer resources and exacerbating disparities in the AI research landscape. Moreover, the escalating energy consumption associated with these computation-intensive models leads to serious environmental concerns (Lacoste et al., 2019; Schwartz et al., 2020; Henderson et al., 2020; Strubell et al., 2020, *inter alia*).

Therefore, building more efficient models for AI systems has become a pressing challenge, drawing widespread attention from the commu-

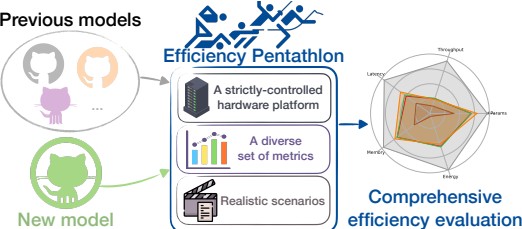

Figure 1: By submitting to Pentathlon, practitioners can compare their models against all previous submissions on identical hardware, eliminating the need to re-implement previous works and substantially reducing the workloads for fair comparisons. Models are evaluated in four realistic scenarios designed to mirror real-world applications. Our platform evaluates the submissions with five crucial efficiency metrics including throughput, latency, memory overhead, # parameters, and energy consumption, hence the name **Penta**thlon.

nity (Reddi et al., 2020; Tay et al., 2020; Treviso et al., 2022; Liu et al., 2022; Yao et al., 2022; Fu et al., 2023b, *inter alia*). However, a lack of standardized evaluation protocols makes it challenging to measure the progress in efficiency improvements and obstructs the efforts in developing more

efficient models. In many cases, models are evaluated in scenarios that hardly reflect the deployment of machine learning models in practice (Henderson et al., 2020). Moreover, some widely-adopted efficiency metrics such as FLOPs often poorly correlate with models' real-world efficiency performance (Dehghani et al., 2022; Fernandez et al., 2023).

The issue is exacerbated by several practical challenges. For instance, hardware is a critical confounding factor in efficiency comparisons, but is very challenging to control in practice, due to disparate levels of hardware accessibility across institutions. Consequently, this leads to disconnections between efficiency improvements in research and tangible progress in practice. There is a pressing need for a standardized efficiency evaluation framework.

To address these challenges, we present Pentathlon. It is designed to establish a standardized platform for evaluating the *inference* efficiency of AI models. As shown by Patterson et al. (2022) and Wu et al. (2022a), inference accounts for over 60% of energy consumption in real-world machine learning workloads. Pentathlon aims to provide comprehensive and realistic evaluation of efficiency, and offer the community a platform to make fair comparisons in a strictly controlled environment. To achieve this, we make several key design choices:

- **Controlled hardware environment** (§2.1): hosted by a dedicated server, Pentathlon provides a centralized platform with a strictly controlled hardware environment. This removes the necessity for practitioners to reproduce previous works on their own hardware for fair comparisons and allows easy comparisons with models previously evaluated on Pentathlon using identical hardware. Moreover, it allows us to use power monitoring devices to accurately measure the energy consumption during models' inference, which was previously impossible.
- **Realistic scenarios** (§2.2): It evaluates models under various scenarios specifically designed to mirror real-world deployment contexts, allowing different approaches to batching input instances, aiming to bridge the gap between research context and practical applications.
- **Comprehensive metrics** (§2.3): Pentathlon evaluates models with five crucial metrics, including throughput, latency, memory overhead, the number of parameters, and energy consumption, hence the name **Penta**thlon. This provides a more holistic understanding of a model's efficiency.
- **Flexibility** (§2.4) Pentathlon is flexible by design and can be seamlessly integrated into any codebase. Although we focus on natural language processing (NLP) models in this paper, Pentathlon can be easily extended to other fields. Finally, the Pentathlon code is easily extended for use with a different server for users who have their own hardware environments.

Pentathlon will be ready to accept submissions after the reviewing period. As we demonstrate in the experiments (§3), Pentathlon can present fresh insights by comparing several established model architectures across a variety of tasks: machine translation, mathematical reasoning, and text classification. In particular, the comprehensive evaluation offered by Pentathlon highlights the effectiveness of quantization in large models. Furthermore, Pentathlon's energy evaluation component reveals new perspectives on the models' energy consumption during inference.

We envision that by offering standardized efficiency evaluation, Pentathlon will stimulate the development of more efficient models and foster a deeper awareness of the computational costs of AI research, and accelerate progress on reducing them.

## 2   🏊🚴🏃 EFFICIENCY PENTATHLON

This section discusses the current challenges in efficiency evaluation and outlines the design choices we adopted in Pentathlon to effectively address them.

### 2.1   CONTROLLING THE HARDWARE FOR FAIR EFFICIENCY COMPARISONS

The hardware stands as a critical confounding factor when comparing efficiency, and can significantly influence the conclusions of such comparisons. As demonstrated by several recent studies, the trends in efficiency comparisons can vary substantially when different accelerators are used (Peng et al., 2021; Kasai et al., 2021a; Wu et al., 2022b; Wang et al., 2020, *inter alia*). Compounding this issue is the practical difficulty in controlling for hardware, primarily because access to hardware platforms often varies among institutions. This is a major obstacle for fair efficiency comparisons. Even with

publicly available implementations, practitioners often need to adapt these to their own hardware environments to ensure fair comparisons.

**Our approach.** Pentathlon aims to stimulate algorithmic innovations that can generalize across different hardware. Therefore we control for hardware while conducting efficiency comparisons and offer a varied selection of hardware to simulate different use cases. Pentathlon is hosted with a dedicated in-house server. Participants can submit their models' code and checkpoints to our server through an easy-to-use tool that we provide (§2.4). This ensures that all models evaluated using Pentathlon use an identical hardware environment, guaranteeing fair comparisons. By requiring code submission Pentathlon helps improve transparency. The specific implementation choices for each submission, such as data IO and padding, will be thoroughly documented. This is appealing because it helps disentangle the efficiency gains due to *algorithmic innovations* from those achieved by better implementations that can equally benefit all models. Further, a dedicated in-house server allows us to measure energy consumption, which would otherwise be very challenging to incorporate (§2.3).

The hosting machine of Pentathlon has two NVIDIA RTX 8000 GPUs, two Intel Xeon Ice Lake Gold 6348 28-Core CPUs, and 1TB DDR4 memory. It supports evaluation using GPUs and CPUs, and CPUs only. To accurately measure each submission's efficiency without interference, we have implemented a scheduler on the server. This ensures that only one inference workload is running at any given time. In Pentathlon, the efficiency measurement begins when the model has been loaded and is ready for predictions, excluding the overhead associated with both model and data loading.

## 2.2 REALISTIC EVALUATION SCENARIOS DESIGNED TO EMULATE REAL-WORLD APPLICATIONS

| Scenarios | Acc. | TP. | Latency | Mem. | Energy & $CO_2$ | BSZ | Online |
|---|---|---|---|---|---|---|---|
| **Fixed batching** | ✓ | ✓ | ✓ | ✓ | ✓ | User specified | ✓ |
| **Poisson batching** | ✗ | ✓ | ✓ | ✓ | ✓ | Random | ✓ |
| **Single stream** | ✗ | ✗ | ✓ | ✓ | ✓ | 1 | ✓ |
| **Offline** | ✗ | ✓ | ✗ | ✓ | ✓ | User specified | ✗ |

Table 1: Four evaluation scenarios and the metrics they focus on. Acc.: accuracy, TP.: throughput, Mem.: memory. In the three online scenarios, Pentathlon interfaces with the submitted model via standard input/output (`stdio`), providing inputs and capturing outputs in real-time. Rearrangement of instance order is prohibited in these scenarios. In the offline scenario, the model is given immediate access to all evaluation instances via a file, enabling techniques such as sorting by lengths.

NLP systems are deployed across a broad range of practical applications, each with its unique requirements for efficiency. Consider, for instance, an online search engine. The arrivals of users' queries are unpredictable, and so is the model's inference batch size. An AI assistant operating on a smartphone typically processes one request at a time, while an offline translation system translating an entire book must use large batch sizes to prioritize maximizing throughput. These practical scenarios are rarely reflected by conventional efficiency evaluations in the research context, where models are typically assessed with a fixed batch size. Such disparity underscores the pressing need for evaluation protocols that better reflect real-world deployments.

**Our approach.** Inspired by Reddi et al. (2020), we include four distinct evaluation scenarios to provide a comprehensive evaluation of NLP models in a variety of realistic settings:

- **Fixed batching.** The evaluation data is first randomly shuffled before being grouped into batches of a user-specified `batch-size`. This setting is intended to mimic typical research experimental settings. We defer to the users choosing optimal batch sizes for their models.
- **Poisson batching** is similar to the fixed batching scenario, but the size of each batch is randomly drawn from a Poisson distribution with a mean of `batch-size`: `batch-size`$_{\text{Pois}} \sim$ Pois(`batch-size`). This setup aims to simulate an online service where the volume of requests is unpredictable but the average can be estimated.
- **Single stream** randomly shuffles the evaluation instances and uses a batch size of one, reflecting the applications processing one request at a time.

- **Offline:** In this scenario, the model has immediate access to the entire evaluation dataset, enabling techniques such as sorting the inputs by length or adaptive batching to enhance throughput and memory efficiency. This scenario reflects large-scale, offline tasks.

These varied evaluation scenarios are designed to highlight the strengths and weaknesses of different models in diverse deployment contexts.

## 2.3 A Diverse Set of Metrics for Comprehensive Efficiency Evaluation

AI systems' efficiency in practical contexts is multifaceted and can hardly be adequately represented by any single metric. Different use cases prioritize different efficiency aspects. For example, a model deployed on mobile devices prioritizes energy efficiency, an offline model requires optimal throughput, while an online service model demands low latency. However, the widely-used metrics often fail to show strong correlations with these diverse practical aspects of efficiency. Take, for instance, the number of floating point number operations (FLOPs) a model takes for performing a workload. It has become a standard efficiency metric partly due to its hardware and implementation-agnostic nature, highlighting the algorithmic advancements in model efficiency (Schwartz et al., 2020). Yet recent research has cast doubt on its relevance, showing that it is a poor indicator of many practical metrics including throughput, latency, and energy consumption (Henderson et al., 2020). Even for models sharing similar architectures and numbers of parameters, their energy efficiency can diverge significantly under identical workloads, partly due to specific deep learning operations they are implemented with (Cao et al., 2021).

This highlights the limitations of conventional evaluation protocols, which risk oversimplifying efficiency comparisons by attempting to encapsulate performance in a single measure. Instead, we propose a more comprehensive approach that considers a diverse suite of metrics. It more accurately reflects the multifaceted nature of efficiency in AI models.

**Our approach.** Our benchmark's suite of evaluation metrics includes the following:

- **Throughput** measures the volume of data a system can process in a unit of time. We measure throughput with instances/s; for tasks that require generating text, we also consider words/s.
- **Latency**, in milliseconds. It quantifies the delay between the system receiving a user request and providing a response. Complementing throughput, it's especially critical in real-time applications, such as smartphone-based AI assistants.
- **Memory overhead**, in GiB, provides insight into a system's applicability in low-resource settings, where available memory can be a bottleneck. In resource-abundant settings, lower memory overhead allows larger batch sizes during inference, improving metrics such as throughput. Our benchmark measures maximum CPU and GPU (if applicable) memory consumption.
- **Energy consumption and carbon footprint.** The energy overhead of a system, measured in W·h, indicates its suitability for battery-powered devices. Combined with carbon intensity data, it can also assess a model's carbon footprint in terms of the amount of $CO_2$ emissions, providing an environmental impact comparison for models deployed in practice. We provide more details about measuring energy consumption in §2.3.1.
- **Model size**, measured in the number of parameters, serves as an indicator of models' storage overhead, and often correlates with its memory overhead.

Our approach provides a holistic view of model efficiency, with each focusing on specific application contexts, allowing practitioners to select efficient methods catered to their applications.

### 2.3.1 Challenges in Measuring Energy and our Solution

While most of the metrics above can be measured with existing tools, accurately measuring energy presents unique challenges, primarily due to the lack of established software for this purpose. Although CUDA offers toolkits to measure GPU power, the power usage of CPUs, DRAM, and disks is only accessible on specific types hardware and requires root access (Khan et al., 2018).

Many existing methods estimate energy consumption for *training* using GPU energy alone (Luccioni et al., 2022; Liang et al., 2022a). However, as we will demonstrate in the experiments, this is not suitable for our purposes for two primary reasons. First, it excludes energy comparisons of models running on CPUs, which our study aims to explore. Second, inference tasks by nature entail more frequent data IO, imposing more workloads on CPUs, DRAM, disks, etc., compared to training. In

our experiments, they account for more than 60% of energy consumption—-a significant increase compared to previous estimates for training (Dodge et al., 2022). Therefore, it is essential to measure not only GPU energy but the total energy consumed by the entire machine accurately.

To this end, we use an energy-monitoring device to measure the power consumption.[1] This data, in conjunction with the model's run time, can be used to calculate the model's energy consumption. Physically connected to the host machine's power cables, this device's sensors provide accurate real-time power usage data. According to the manufacturer, the error rate is $\pm 1.2\%$.

The power consumption is calculated by subtracting the host machine's idling power from the meter reading during an inference run. To calculate the carbon emissions, we use the carbon intensity data provided by Schmidt et al. (2022) based on the geographical location and time of the day.

### 2.4 Ensuring Flexibility in Pentathlon

Requiring code and checkpoint submission imposes additional implementation effort from participants, a tradeoff we believe is worthwhile for achieving fair comparisons on a strictly-controlled hardware platform. Recognizing from past benchmark efforts that this might discourage practitioners from participating, we have made a concerted effort to ensure that Pentathlon can be easily integrated into existing code bases and to streamline the submission process. Additionally, our code is written to support use with different servers beyond our dedicated instance, which allows users to provide their own hardware environments, if they wish. Pentathlon supports all tasks and datasets from Hugging Face, and can be easily extend to many other tasks and research fields.

**Accommodating diverse software frameworks.** We aim to encourage wide participation and ensure our platform is accessible to practitioners accustomed to various software infrastructures. Therefore, Pentathlon makes no assumption about the submission's deep learning framework (if a deep learning model is used at all) or the programming language it's implemented in. We require that every submission: (1) Include a GitHub repository containing the code and listing dependencies (this repository does not need to be public); (2) Interface the model to read inputs from `stdin` and write outputs to `stdout`;[2] (3) Implement the necessary tools to download the model checkpoint for evaluation. We provide detailed instructions and examples to guide practitioners through this process. Based on our internal testing, learning to integrate Pentathlon into an existing codebase and submitting it to our server for evaluation takes a participant less than one hour; and an onward submission takes a single command line. Furthermore, Pentathlon can serve as a standalone tool for preparing the submission and providing basic efficiency metrics.

In providing abstractions around the evaluation interface, we limit assumptions made around the underlying system implementation and allow for the installation of user dependencies as needed. This enables support for a diversity of backend frameworks and runtimes as the user is not constrained to a single deep learning framework or data format. For example, Pentathlon allows users to use both research frameworks (e.g., eager execution PyTorch and TensorFlow 2.0) as well as specialized inference runtimes (e.g., ONNX Runtime, TVM, and TensorRT). The additional flexibility provided by this format allows Pentathlon to remain accessible to researchers familiar with a particular framework, while also enabling the exploration of different means of increasing overall *end-to-end efficiency* of the machine learning system that is available in deployment settings. This design allows users to evaluate efficiency gains from improving different aspects of the overall system, such as those obtained from optimizing the model architectures or from utilizing faster software frameworks.

## 3 Experiments

We use Pentathlon to benchmark several established models on a variety of tasks, examining: machine translation (WMT14 DE-EN), mathematical reasoning (GSM8K), and classification (RAFT). We examine models from both the encoder-decoder (e.g. FLAN-T5, MBART) and decoder-only (e.g.

---

[1]We use an emonTx V4 for power consumption measurement: https://shop.openenergymonitor.com/single-phase-6-channel-energy-monitoring-emontx-v4/.

[2]We provide a Python tool for this `stdio` interaction. Users can implement their own interfaces if they decide to use other programming languages.

LLaMa) families of transformer architectures. In the interest of space, we refer the readers to the appendices for experiments on text classification with the RAFT dataset (Alex et al., 2021).

## 3.1 MACHINE TRANSLATION

Improving the efficiency of machine translation (MT) and text generation models has gained significant momentum. A growing number of recent workshops and shared tasks have held dedicated efficiency tracks (Birch et al., 2018; Hayashi et al., 2019; Heafield et al., 2020; Akhbardeh et al., 2021; Kocmi et al., 2022, *inter alia*). Aligned with this goal, we seek to contribute to this ongoing effort. To this end, our initial experiments with Pentathlon focus on machine translation.

**Dataset and setting.**  We present results for WMT14 DE-EN (Bojar et al., 2014), a well-studied dataset that is selected as the testbed in the efficiency tracks of two recent WMT workshops (Akhbardeh et al., 2021; Kocmi et al., 2022). Pentathlon already supports many other MT and text generation datasets, and can be easily extended to more. We focus on DE->EN translation here; additional results with EN->DE are available in the Appendices.

Balancing the inference wall clock time and accurately measuring the efficiency, we use different numbers of evaluating instances across the four scenarios. For WMT14 DE-EN:

- **Fixed batching** uses the full test set of 3,002 instances. It also measures the translation quality using SacreBLEU (Post, 2018).
- **Poisson batching** randomly draws 4,000 instances (with replacement) from the test set.
- In the **single stream** scenario, 1,000 randomly selected test instances are used.
- Differently from others, the **offline** scenario randomly selects 8,000 instances from the *training* data.[3] We ensure that the selected instances have an average length matching that of the test set.

Controlling for the random seed, all models are evaluated on the same set of instances in the same order, and identical batch sizes in the Poisson batching scenario. Preliminary experiments indicate that the models' efficiency performance remains consistent across multiple runs. As such, we opt out of conducting multiple rounds of evaluation. All models are evaluated on one RTX8000 GPU, and the inference batch sizes for the fixed batching and offline scenarios are tuned to the allowable maximum for the available GPU hardware.

**Models.** We benchmark the following publicly-available models covering a wide range of sizes:

- **MBART** (Tang et al., 2021): a 610M-parameter-sized Transformer model for multilingual translation. It has two variants, many-to-one (MBART M2O) translates other languages into English, and many-to-many (M2M) can translate between multiple language pairs. We use the **MBART50** variant, originally pre-trained on monolingual corpora in 25 languages, by fine-tuning on parallel corpora in across 50 languages for direct use as a translation engine.
- **M2M100** (Fan et al., 2021): Transformer-based multilingual models for many-to-many translation. We report on two sizes with 418M and 1.2B parameters respectively. The **M2M100** model is trained using parallel corpora (e.g., WMT corpora described above) and mined bitext to enable translation between any two of 100 languages.
- **OPUS** (Tiedemann & Thottingal, 2020): a bilingual Transformer model with 74M parameters for DE->EN translation. The model is trained on OPUS bitext corpora (Tiedemann, 2012).
- **WMT19-Meta** (Ng et al., 2019): a DE->EN Transformer model with 314M parameters. This system won the WMT19 task on German to English news translation (Barrault et al., 2019).
- **WMT21-Meta** (Tran et al., 2021): a M2O Transformer model with 4.7B parameters. Unlike **WMT19-Meta**, this model is multilingual and trained on data from all languages for the WMT 2021 shared task.Training data is a mixture of parallel corpora, monolingual corpora and mined bitext. This multilingual system ranked high in several WMT21 news translation tasks (Akhbardeh et al., 2021). We refer to Tran et al. (2021) for complete details.

We evaluate using PyTorch with both full precision (FP32) and half precision (FP16), to study the effect of quantization. In our preliminary experiments, we found that employing more aggressive quantization techniques such as 8-bit and 4-bit quantization using naive methods led to severely

---

[3]In this scenario the models are granted immediate access to all instances and can sort them by length. If the instances *were* drawn from the test set, this would result in the artifact that groups duplicates of the same instance in the same batch, which we aim to avoid.

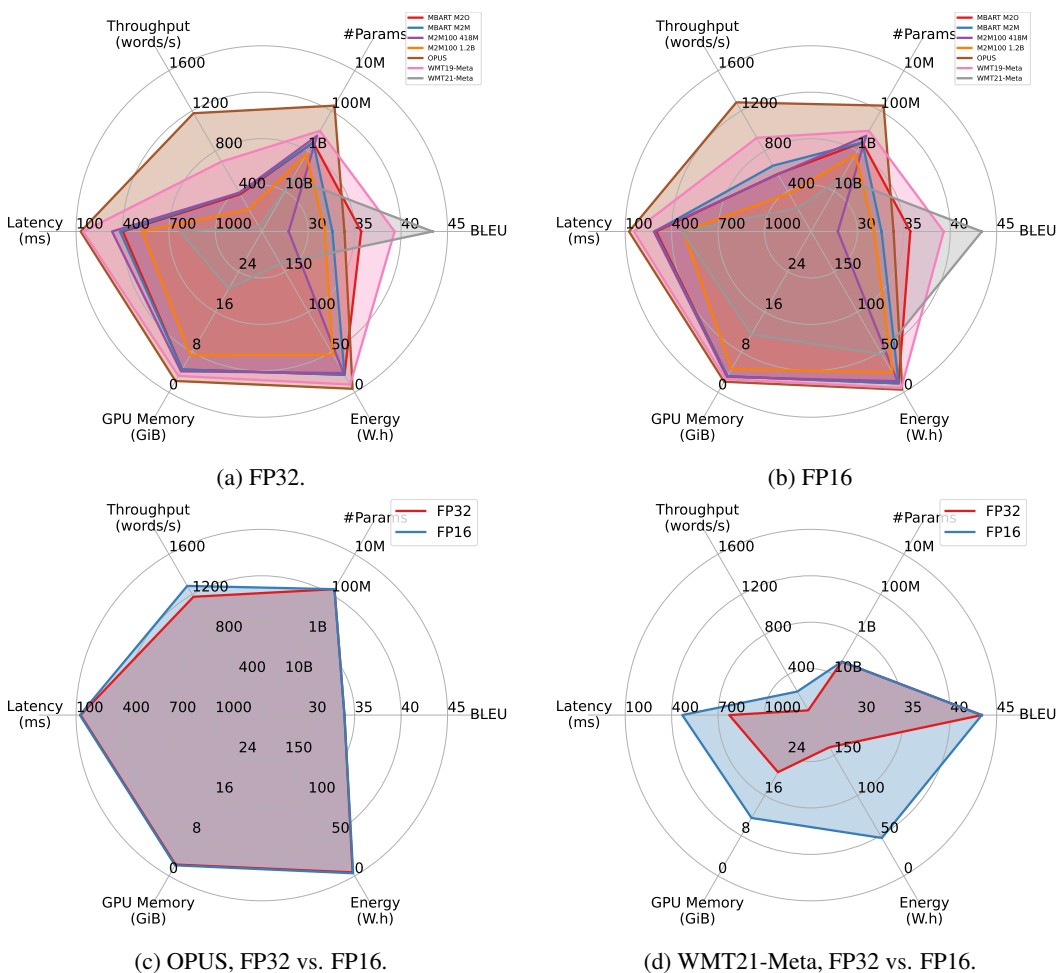

Figure 2: Performance of various models on the WMT14 DE-EN, represented in terms of BLEU scores and a range of efficiency metrics. To more accurately reflect real-world applications, the figures include throughput metrics from the offline scenario, latency and GPU memory metrics from the single stream scenario, and energy metrics from the fixed batching scenario. For all metrics, **outer rings indicate better performance**. #Params is presented on a logarithmic scale.

compromised translation quality, with the BLEU score dropping to around 1, effectively resulting in a failed translation. All models' implementation and checkpoints are available on Hugging Face.

**Results.** Figure 2 summarizes the efficiency performance of different models in on the WMT14 DE-EN dataset, along with their translation quality. Overall, models trained for English translation demonstrated better trade-offs between translation quality and efficiency. Notably, OPUS outperforms the much larger MBART M2M and M2M100 models in both accuracy and all aspects of efficiency, and is the most efficient model among all. Although WMT21-Meta, the largest model considered, provides the highest BLEU score, it takes a substantial hit in efficiency.

Interestingly, despite being more than four times larger, WMT19-Meta achieves efficiency performance comparable to OPUS in latency, memory overhead, and energy consumption, and significantly outperforms it in terms of BLEU. However, it falls short of OPUS in throughput. This observation confirms that relying on a single efficiency metric risks oversimplifying the complex performance landscape of efficiency in practical applications. Utilizing ONNX Runtime with a CUDA Execution Provider, the models achieve over 20% improvements in latency and throughput in the single-stream scenario, accompanied by a significant reduction in memory and energy overhead. However, less efficiency improvement is observed in other scenarios with larger batch sizes.

**Larger models benefit more from FP16 quantization.** By comparing Figures 2a and 2b, we observe that FP16 quantization improves all models' efficiency performance (except #Params.),

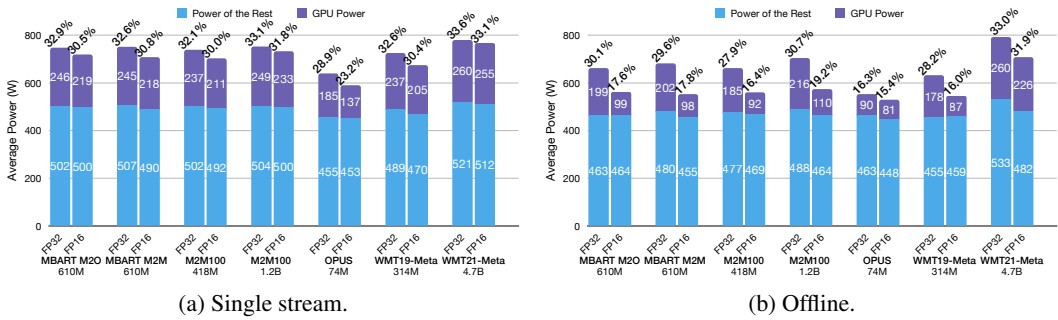

(a) Single stream.  (b) Offline.

Figure 3: Power consumption in Watts across different model inference runs in the single stream (3a) and offline (3b) scenarios. Purple bars indicate the power consumed by the GPU, while the light blue bars represent the power consumption of all other system components, excluding the GPU. The white numbers denote the absolute power consumption values in Watts, while the percentage numbers atop the bars provide the proportion of power consumption that is accounted for by the GPU.

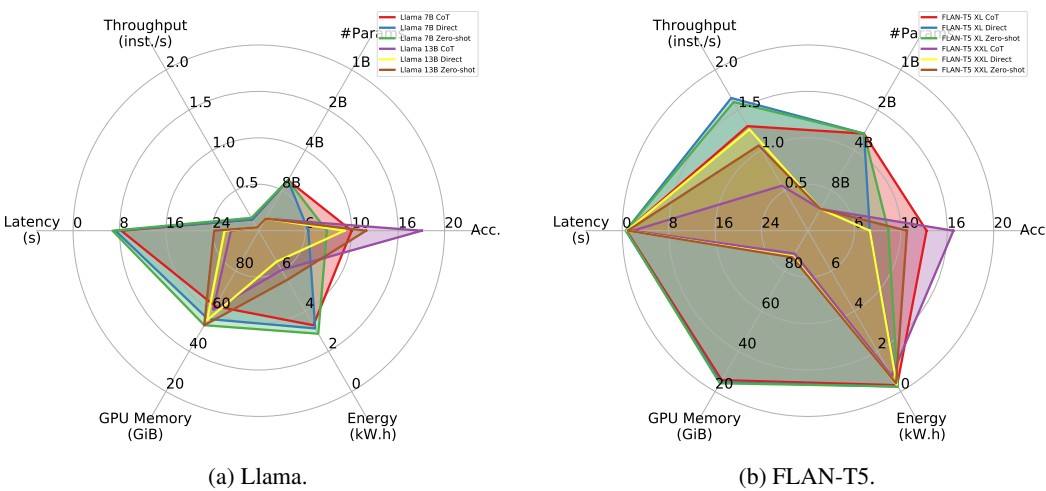

(a) Llama.  (b) FLAN-T5.

Figure 4: In-context learning performance of Llama (left) and FLAN-T5 (right) models on the test split of GSM8K, represented in terms of accuracy and a range of efficiency metrics. All models are evaluated on two NVIDIA RTX 8000 GPUs with FP32 precision.

particularly memory overhead. Larger models appear to benefit more from quantization. As shown in Figures 2c and 2d, while OPUS experiences minimal efficiency gains from quantization apart from increased throughput, WMT21-Meta's efficiency dramatically improves with FP16 quantization, nearly doubling throughput and reducing latency, memory overhead, and energy consumption by half or more. These results highlight the promise of advancing quantization techniques for larger models in order to improve the trade-off between accuracy and efficiency.

**In single-GPU inference, the GPU accounts for only a minor portion of the energy consumption.** This is demonstrated by Figure 3. This experiment uses a single RTX8000 GPU with a maximum power of 260W. The GPU rarely operates at full power, implying that GPU hours, a metric commonly used to gauge training computational overhead (Henderson et al., 2020; Kasai et al., 2021b), is unsuitable for estimating inference GPU energy. Even during the most GPU-intensive runs by the WMT21-Meta model, where it does operate at full capacity, the GPU only accounts for one third of the total machine power. This observation diverges from previous findings on *training*, where GPUs are estimated to constitute around 70% of the energy usage (Dodge et al., 2022). We attribute the difference to the increased memory and disk IO demands during inference, coupled with lower GPU utilization and increased idling time due to smaller compute kernels during inference This suggests that efficiency conclusions drawn from training need careful examination when applied to inference. Interestingly, we observe a correlation between higher GPU power and higher power by other components. We conjecture that this is partially due to the increased fan activities for cooling.

### 3.2 MATHEMATICAL REASONING WITH GSM8K

**Dataset and settings.** We use GSM8K (Cobbe et al., 2021), a mathematical reasoning dataset containing grade school level math word problems. Our implementation follows that of Fu et al. (2023a). All models are evaluated on two NVIDIA RTX 8000 GPUs, with FP32 precision. The maximum output length is 256.

We consider three prompting settings: **CoT** uses the 8-shot chain-of-thought demonstration crafted by Wei et al. (2022); **Direct** prompts with the same set of instances, but without reasoning chains; **Zero shot** does *not* use any in-context demonstrations and prompts with the test instance.

We compare the following publicly-available LLMs:

- **FLAN-T5** (Chung et al., 2022): an encoder-decoder Transformer model that has undergone large-scale instruction finetuning. We consider both the XL and XXL sizes, with 3B and 11B parameters respectively. We use the implementations by Hugging Face.
- **Llama** (Touvron et al., 2023): a suite of LLMs released by Meta. We compare both the 7B and 13B sized models. We compare both the Hugging Face implementation.

**Results.** Figure 4 compares the models' accuracy and efficiency performance. We conjecture that the strong performance by FLAN-T5 can be partly attributed to its instruction finetuning. Interestingly, although FLAN-T5 XXL and Llama-13B are of similar sizes, the former is more efficient in terms of throughput, latency, GPU memory, and energy consumption. This is because Llama tends to generate much longer outputs. Overall, FLAN-T5 XXL achieves a strong trade-off between accuracy and efficiency, outperforming Llama models of a similar scale.

## 4 RELATED WORK

There is growing interest in putting efficiency in NLP benchmarks. Dynabench (Kiela et al., 2021) and Dynaboard (Ma et al., 2021) concentrate on dynamic dataset creation and model assessment, incorporating efficiency metrics such as throughput and memory, alongside fairness and robustness HELM (Liang et al., 2022b) evaluates language models with seven metrics including efficiency. Though training efficiency in HELM covers energy, carbon, and wallclock time, the inference efficiency in this benchmark only measures inference runtime, and the energy and carbon footprint are only roughly estimated. HULK (Zhou et al., 2021) evaluates energy efficiency as a proxy of time and cost, while Pentathlon evaluates multiple different efficiency metrics in a realistic way. Long-Range Arena (Tay et al., 2021) builds a set of synthesized tasks to evaluate the long-range capabilities of NLP models in terms of generalization and computational efficiency including speed and memory footprint. Another line of work has studied application- or task-specific efficiency such as trade-offs between accuracy and energy consumption for long context NLP models (Ang et al., 2022), inference energy competition for models on SuperGLUE (Wang & Wolf, 2020) or storage efficiency for open domain question answering (Min et al., 2021). Most related to Pentathlon, MLPerf targets inference efficiency across various real-world scenarios (Reddi et al., 2020; Banbury et al.; Mattson et al., 2020). While MLPerf aims to stimulate building more efficient hardware platforms, Pentathlon incentivizes algorithmic innovations, controlling the hardware. Hosted on an in-house machine, Pentathlon can accurately measure inference energy consumption, which was impossible for previous benchmark efforts.

## 5 CONCLUSIONS

We present Pentathlon, a benchmark for holistic and realistic evaluation of inference efficiency. Pentathlon targets multiple aspects of efficiency including latency, throughput, memory overhead, number of parameters, and energy consumption, on a strictly-controlled hardware platform. Integrating evaluation with Pentathlon is seamless and can drastically reduce the workload to make fair and reproducible efficiency comparisons. Pentathlon offers both testing in real-world application scenarios and a standardized platform for comparison between any two submissions. We establish this tool for NLP models but offer flexible extensions to additional tasks and scenarios. We envision Pentathlon to provide a new lens on testing algorithmic innovations by lowering the barrier to entry for evaluating efficiency and characterizing environmental impact of future models.

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

# Appendices

## A  TEXT CLASSIFICATION WITH RAFT

RAFT is a collection of 11 datasets that focus on few-shot text classification in real-world settings. Here we focus on the ADE Corpus V2 (ADE) portion, aiming to classify sentences derived from medical reports as related or unrelated to adverse drug effects. Several baseline models, provided by the authors, were evaluated for efficiency, including:

- **AdaBoost** (Freund & Schapire, 1995): a strong non-neural classifier based on decision trees.
- **BART Zero Shot MNLI**: BART (Lewis et al., 2020) finetuned on the MNLI dataset (Williams et al., 2018). It is used as a zero-shot classifier.
- **GPT-2** (Radford et al., 2018): used as a few-shot classifier with 25 in-context training demonstrations and task-specific instructions.

The implementation for all models is attributed to Alex et al. (2021).[4] At the time of writing, RAFT has not released the gold labels of the test split, and therefore we report the F1 performance by Alex et al. (2021).

**Results.** Figure 5 provides a comparison of the above models with a majority-class baseline (Maj.). AdaBoost, a non-neural model, emerges as a strong competitor in terms of the accuracy-efficiency trade-off. Interestingly, GPT-2, despite having fewer parameters, lags behind BART in terms of throughput, latency, and energy consumption. We believe that this could be due to the in-context few-shot examples, which lead to significantly longer inputs for GPT-2 compared to BART. Nonetheless, GPT-2 manages to achieve the highest F1 score in this experiment.

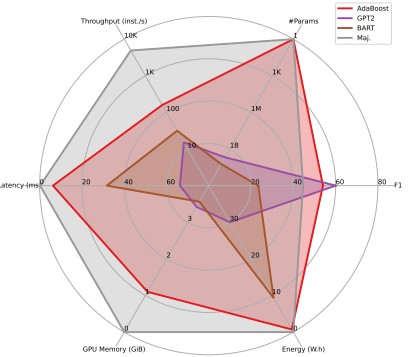

Figure 5: Models' efficiency and accuracy performance on RAFT test data. F1 numbers are due to Alex et al. (2021).

## B  ADDITIONAL EXPERIMENTS WITH MACHINE TRANSLATION

All models' implementation and checkpoints are available on Hugging Face, with the following identifiers:

- MBART50: `facebook/mbart-large-50-many-to-{many, one}-mmt`;
- M2M100: `facebook/m2m100_{418M, 1.2B}`;
- OPUS: `Helsinki-NLP/opus-mt-de-en`;
- WMT19-Meta: `facebook/wmt19-de-en`;
- WMT21-Meta: `facebook/wmt21-dense-24-wide-en-x`.

**Additional FP32 vs. FP16 comparisons.** Figure 6 provides an additional set of comparisons between FP32 and FP16 across various models on WMT14 DE-EN, complementing the results presented in Section 3. The general trends mirror those observed earlier, with larger models benefiting more in terms of efficiency from quantization compared to smaller ones.

**ONNX improves throughput, latency, and energy overhead, at the cost of increased GPU memory overhead.** Pentathlon makes little assumptions on the models' implementation and backend runtime, and allows users to use both eager-execution research frameworks like PyTorch as well as specialized inference runtimes like Open Neural Network Exchange (ONNX). Here we study ONNX's impact on the model's efficiency.

ONNX is a cross-platform static runtime that uses pre-compiled computational graphs. It allows for aggressive, global ahead-of-time compiler optimizations, and can bring substantial latency and throughput improvements in inference settings with small batch size. The readers are referred to

---

[4]`https://github.com/oughtinc/raft-baselines`

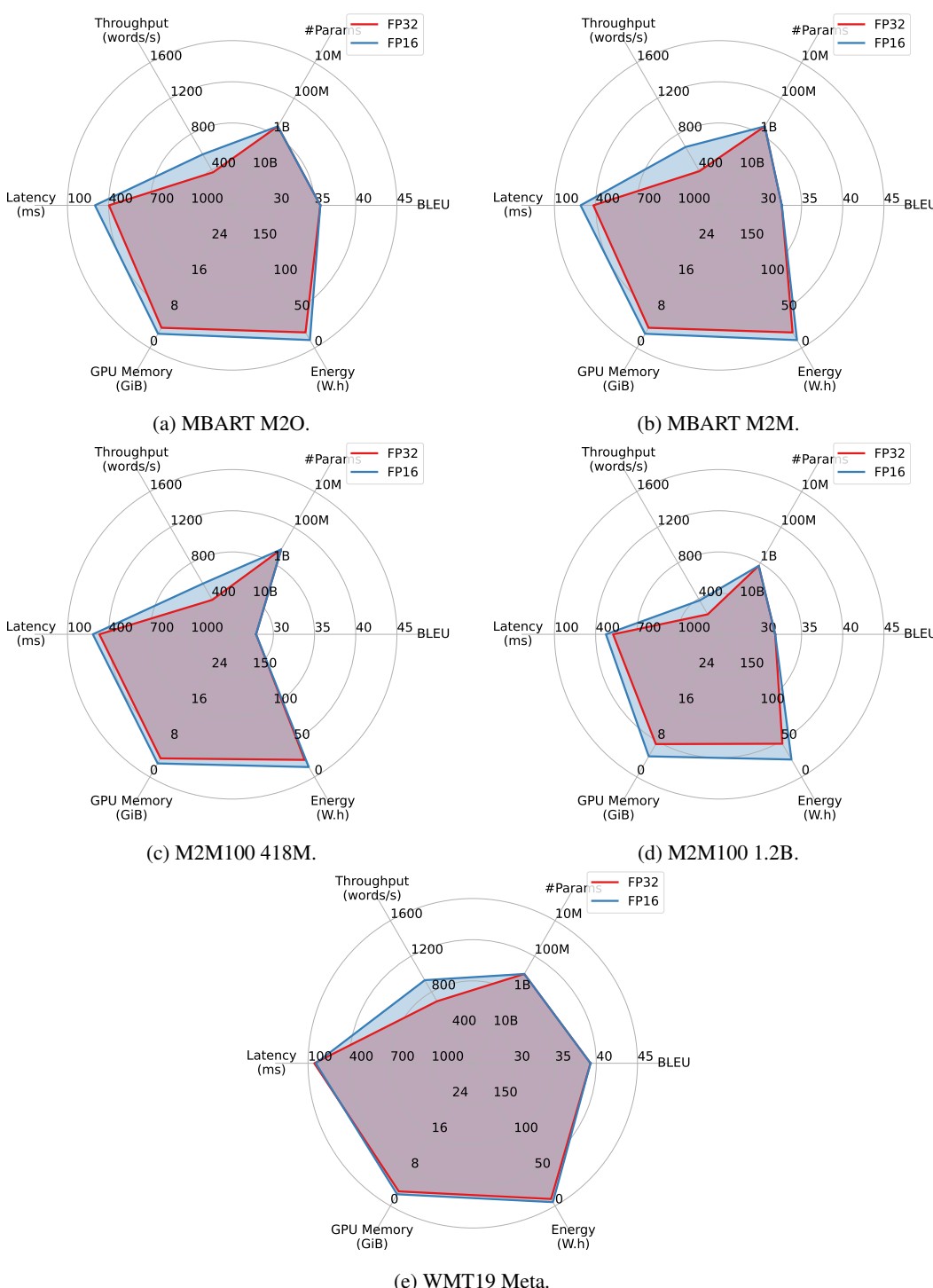

Figure 6: Additional results of various models on the WMT14 DE-EN using FP32 (red) and FP16 (blue). Similarly to Figure 2, the throughput metrics are from the offline scenario, latency and GPU memory metrics from the single stream scenario, and energy metrics from the fixed batching scenario.

https://onnx.ai/ for more details. As of now, ONNX supports conversion from models implemented with PyTorch, Tensorflow, and JAX, enabling us to make direct comparisons between PyTorch implementation and ONNX in our machine translation experiments with WMT14 DE-EN.

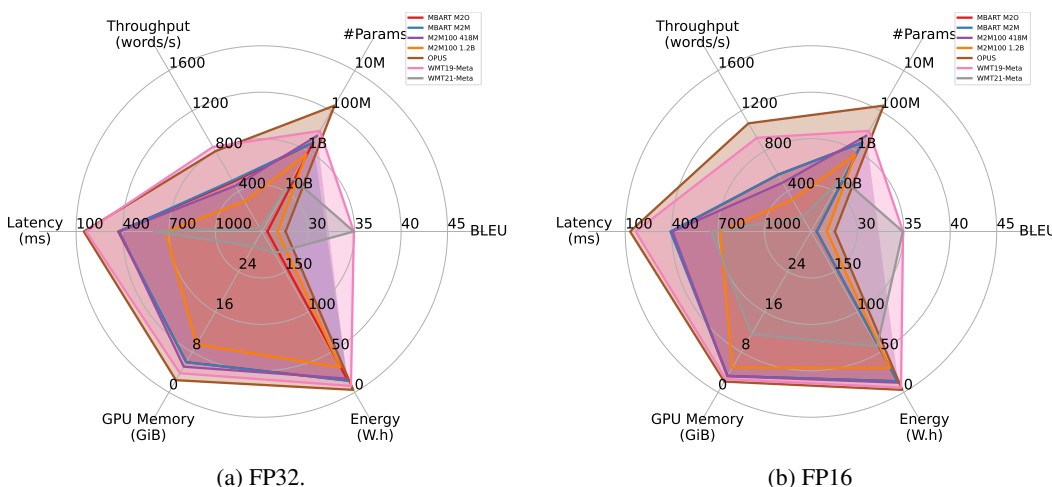

(a) FP32.                                    (b) FP16

Figure 7: Performance of various models on the WMT14 EN-DE. Following Figure 2, the figures include throughput metrics from the offline scenario, latency and GPU memory metrics from the single stream scenario, and energy metrics from the fixed batching scenario. For all metrics, **outer rings indicate better performance**. #Params is presented on a logarithmic scale.

As shown in Figure 8, when comparing five different models in a single-stream scenario using PyTorch and ONNX runtime, ONNX delivers substantial improvements in throughput, latency, and energy overhead, especially for larger models. However, this comes with an increase in GPU memory consumption, which is likely due to the storage of pre-compiled computational graphs on the GPU. WMT19 Meta and WMT21, which utilize the Fully Sharded Data Parallel technique (FSDP; Zhao et al., 2021), are excluded from this experiment due to compatibility challenges with ONNX and FSDP.

Our preliminary experiments find that ONNX brings marginal efficiency improvements in other scenarios that use larger batch sizes, which is consistent with the observation by Fernandez et al. (2023).

**Results on WMT14 EN->DE.**

Figure 7 provides a summary of the efficiency performance of various models on the WMT14 English-to-German (EN->DE) translation task. The results are shown for both FP32 and FP16 models. The observed trends align with those discussed in Section 3.

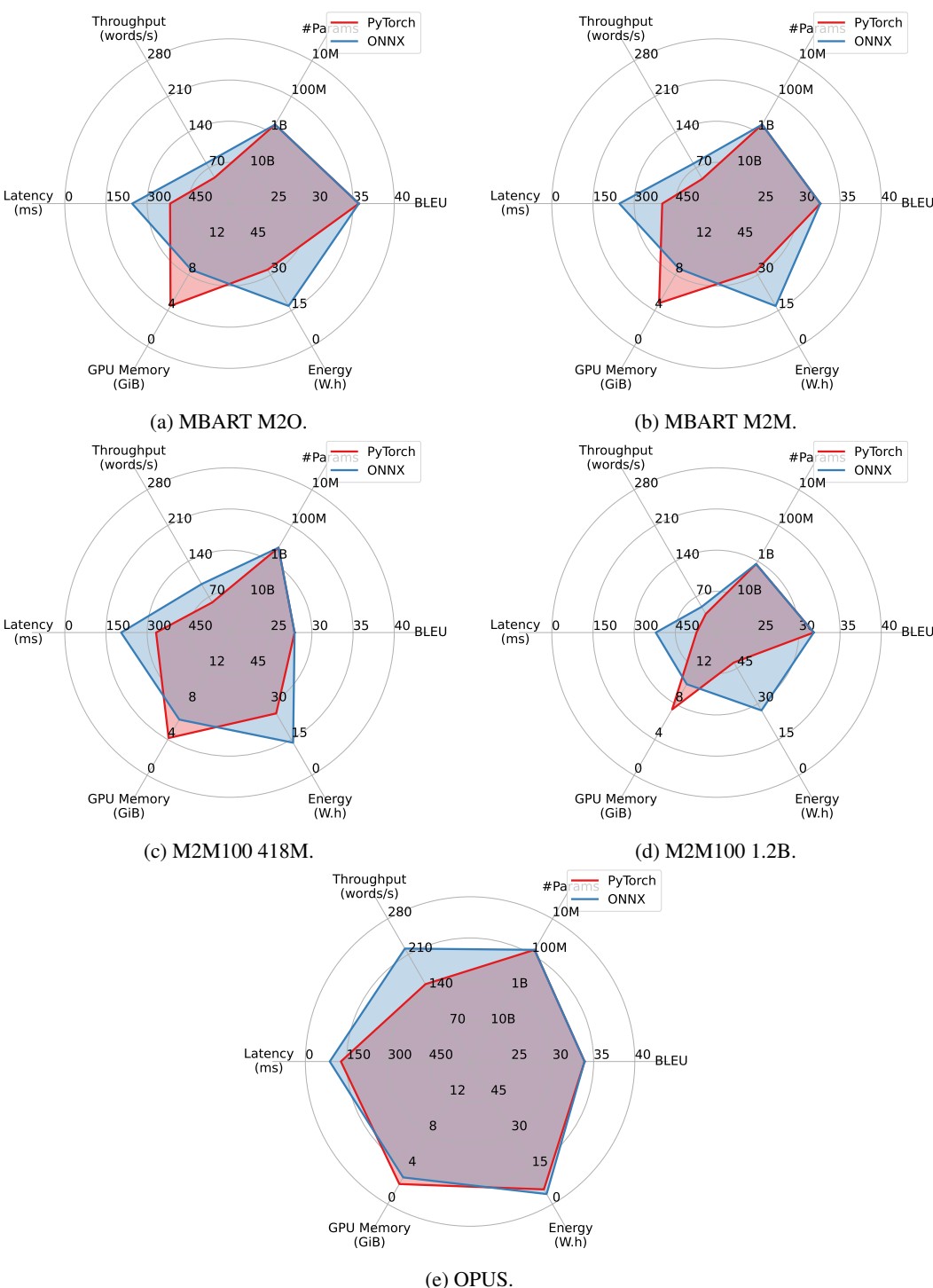

Figure 8: Accuracy and efficiency performance comparisons of five different models while using PyTorch (red) and ONNX (blue) runtime. WMT19 Meta and WMT21 Meta rely on the Fully Sharded Data Parallel (FSDP; Zhao et al., 2021) in their implementation, which complicates their conversions to ONNX, and are therefore not included in this figure. All efficiency metrics are measured in the single-stream scenario; in preliminary experiments, we observe that the efficiency gains from ONNX are marginal in other scenarios, as expected.

