# OpenReview forum: "Efficiency Pentathlon: A Standardized Benchmark for Efficiency Evaluation"
_ICLR.cc/2024/Conference — Submitted to ICLR 2024_

### Official Review · Reviewer_pV3M · 2023-10-30

**Soundness:** 2 fair
**Presentation:** 2 fair
**Contribution:** 3 good
**Rating:** 5
**Confidence:** 3

**Summary:**

Evaluating the model efficiency is an important evaluation aspect for practical applications. The paper claimed that existing metics, such as FLOPs often did not reflect the advantages of the models in real-world applications. So, it proposes efficiency Pentathlon, a benchmark of holistic and realistic evaluation of model efficiency, including five ways: standard hardware enviroment, four distinct evaluation scenarios, diverse metrics for comprehensive efficiency evaluation, a evaluation software library, and flexiable evaluations. The established benchmark contain three NLP tasks and the corresponding datasets (WMT14 DE-EN, GSM8K, RAFT). The evaluation results show that the proposed Pentathlon could drastically reduce the workoad to make fair and reproducible effciency comparisons.

**Strengths:**

1) Efficiency evaluation is very important for model evlaution and seldomly addressed before.
2) The proposed five evaluation aspects are interesting and novel.

**Weaknesses:**

1) The whole paper is not very clear. In the section 2,  the reason that considering the proposed five aspects for effeciency evaluation is not described clearly.
2) The experimental parts are not very suffcient. Only two tasks are selected which makes the results not very convincing.

**Questions:**

1) Table 1 is not very clear. What is the schema meaning in table 1, such as Acc., TP., Latency, Mem., etc. The authors should describe them in the tabel title.
2) What did only three kinds of NLP tasks are selected? I concern what are the results in other NLP tasks.
3) Why this benchmarks could be trusted and applied when evaluting the real-applications. The authors should prove the advantages of the proposed benchmarks and plantforms.

---

> ### Author Response · Authors · 2023-11-23
>
> We thank the reviewer for their feedback. We are glad that the reviewer finds efficiency evaluation important and some of our design choices interesting and novel.
>
> > The whole paper is not very clear. In the section 2, the reason that considering the proposed five aspects for efficiency evaluation is not described clearly.
>
> We believe that using multiple metrics to evaluate a model's efficiency is crucial for a comprehensive understanding of its performance and suitability for various applications. It can also help understand the tradeoff among these aspects and help make better decisions while deploying the models. Each metric provides a different perspective
> - Latency is crucial for real-time applications, such interactive systems like chatbots,  and real-time translation systems, where quick response is critical for the user’s experience.
> - Throughput is important in scenarios where the model needs to process large volumes of data, e.g., batched processing of many scientific papers or news articles.
> - Memory overhead can reveal whether a model might not be suitable for deployment on devices with limited memory, such as mobile phones or IoT devices.
> - Energy consumption is important for justifying the model’s suitability for battery-powered devices or in situations where energy efficiency is a priority.
> - Model size  affects its portability and the speed of its deployment. Smaller models are easier to distribute and can be deployed on devices with limited storage capacity.
>
> We will highlight these motivations in the revision.
>
> > The experimental parts are not very suffcient. Only two tasks are selected which makes the results not very convincing.
>
> > What did only three kinds of NLP tasks are selected? I concern what are the results in other NLP tasks.
>
> Our decision of experimenting with three tasks (instead of two) aims to demonstrate the Pentathlon’s versatility supporting a variety of architectures and settings.  Please refer to the general response for further details.
>
> > Table 1 is not very clear. What is the schema meaning in table 1, such as Acc., TP., Latency, Mem., etc. The authors should describe them in the tabel title.
>
> We chose to use abbreviations in the interest of space, which are described in the caption of Table 1. Acc.: accuracy, TP.: throughput; Mem.: memory overhead.

---

### Official Review · Reviewer_uM5U · 2023-11-01

**Soundness:** 2 fair
**Presentation:** 3 good
**Contribution:** 2 fair
**Rating:** 3
**Confidence:** 4

**Summary:**

This paper presents Pentathlon, a benchmark for holistic and realistic evaluation of model efficiency. The benchmark offers a strictly controlled hardware platform and incorporates metrics to measure efficiency, including latency, throughput, memory overhead, number of parameters, and energy consumption. The authors also provide a software library that can seamlessly integrate into any codebase and enable evaluation.

**Strengths:**

- The paper introduces a standardized and centralized evaluation platform, which can reduce the workload to make fair and reproducible efficiency comparisons and stimulate algorithmic innovations in building efficient models.

**Weaknesses:**

- The paper is more like a technical report, which may suit a benchmark or industry track. It would be great if the authors could provide additional scientific findings and conclusions through the evaluations. This work has provided a relatively comprehensive and mature evaluation benchmark. With more inspiring and interesting findings and conclusions based on the evaluations, the paper would be more valuable to the community.

- There are many datasets designed for large language model evaluation. Using classical tasks (e.g., machine translation, mathematical reasoning, and classification) makes the experiments less convincing.

**Questions:**

See weaknesses.

---

> ### Author Response · Authors · 2023-11-23
>
> > The paper is more like a technical report, which may suit a benchmark or industry track. It would be great if the authors could provide additional scientific findings and conclusions through the evaluations. This work has provided a relatively comprehensive and mature evaluation benchmark. With more inspiring and interesting findings and conclusions based on the evaluations, the paper would be more valuable to the community.
>
> We agree with the reviewer that this submission fits the datasets and benchmarks track of ICLR 24, the exact track that this paper is submitted to: https://iclr.cc/Conferences/2024/CallForPapers
>
> We believe that the value of this work lies in providing the community with a platform that can help discover interesting findings and conclusions about the efficiency of machine learning models.
>
>
> > There are many datasets designed for large language model evaluation. Using classical tasks (e.g., machine translation, mathematical reasoning, and classification) makes the experiments less convincing.
>
> Our decision to experiment with machine translation, classification, and math word problems aims to demonstrate the Pentathlon’s versatility supporting a variety of architectures and settings.  Please refer to the general response for further details.

---

### Official Review · Reviewer_5UHy · 2023-11-01

**Soundness:** 3 good
**Presentation:** 4 excellent
**Contribution:** 3 good
**Rating:** 5
**Confidence:** 3

**Summary:**

This paper introduces `Pentathlon`, a benchmark created for the comprehensive and realistic evaluation of model inference efficiency. Pentathlon provides a strictly controlled hardware platform, including GPUs and CPUs, and incorporates a suite of metrics targeting different aspects of efficiency, including latency, throughput, memory overhead, parameter count, and energy consumption. As a standardized and centralized evaluation platform, Pentathlon aims to significantly reduce the workload required for fair and reproducible efficiency comparisons. While its initial focus is on natural language processing (NLP) models, Pentathlon is designed to be flexibly extended to other fields.

**Strengths:**

-   Clarity: The paper is exceptionally well-written, offering a comprehensive presentation of the Pentathlon benchmark suite. The authors provide a detailed explanation of its design, which emphasizes equitable model comparisons and incorporates testing settings for both CPUs and GPUs.

-   Thoughtful Metric Selection: Pentathlon's use of five carefully chosen evaluation metrics addresses critical properties of models, ensuring that the benchmark accurately assesses key aspects of efficiency.

-   Visual Aid: The inclusion of radar charts is a notable advantage, as they effectively illustrate the strengths and weaknesses of models, making it easier for readers to comprehend the benchmark's findings.

-   Centralized Benchmarking: While the concept of centralized benchmarking isn't entirely novel, it remains highly valuable for gaining a deeper understanding of the diverse impacts algorithms have on model efficiency. Pentathlon offers a structured and standardized approach to this essential process.

-   Realistic Workloads: The authors' meticulous design of workloads to mirror batching scenarios and real-world service loads enhances the reliability of the benchmark's results, ensuring they are more reflective of practical use cases.

**Weaknesses:**

1.  Hardware Flexibility: While the authors have outlined CPU and GPU settings, it remains unclear whether the benchmark suite can easily accommodate other hardware platforms. Given the growing popularity of new platforms like Metal, ROCm, Mali, and Vulkan, it's essential to address the adaptability of Pentathlon to ensure it remains relevant and applicable to diverse hardware configurations. Moreover, certain models, such as Llama-70b, may require multiple high-end GPUs like A100 or H100 for distributed inference, highlighting the need for flexibility in hardware options.

2.  Software Environment Assumptions: The paper primarily focuses on a specific software environment, potentially overlooking the fact that various software stacks, such as TVM and Cutlass, may require an additional step called tuning. This tuning phase optimizes the compilation stack for the given hardware, which can significantly improve model performance. However, the tuning process itself may not always be efficient and can be time-consuming. It's crucial to consider these software-related aspects for a more comprehensive evaluation.

3.  Controlled Hardware vs. Cloud-Based Platforms: While the controlled hardware setting provides fairness and accuracy, it may not fully cater to researchers who heavily rely on cloud-based platforms like AWS, Azure, or Google Cloud. Many recent large language models (LLMs) are built and deployed on cloud platforms, and their efficiency and latency results may significantly differ from those obtained in a controlled environment. To make Pentathlon more applicable to a wider range of real-world scenarios, consideration could be given to extending the benchmarking to include cloud-based machines and their specific challenges.

**Questions:**

1.  What level of effort is required to expand Pentathlon to accommodate a new hardware platform or incorporate a new model into the benchmark?
2.  Beyond the BLEU score, are there additional metrics available within Pentathlon to assess model quality, such as perplexity or other relevant NLP-specific metrics?
3.  How can you distinguish the impact of "algorithmic innovations" from other efficiency-related factors in the Pentathlon benchmark?

---

> ### Author Response · Authors · 2023-11-23
>
> We thank reviewer 5UHy for their constructive feedback. It is encouraging to hear that they found our work well-written and appreciated many of our design choices.
>
> > Hardware Flexibility
>
> We appreciate the reviewer for raising the valid concern that controlling the hardware leads to less flexibility. In preliminary experiments, we found it possible to simulate mobile phone platforms by connecting to our Linux-host an Arm chip device. Some on-device applications can be simulated by running our platform on the NVIDIA Jeston module: https://developer.nvidia.com/embedded/jetson-tx2
>
> We acknowledge the challenge of continuously supporting new hardware platforms, which is beyond the capabilities of the authors. Trying to address this, Efficiency Pentathlon is designed to be flexible, and can be used as a standalone software without our machine. It is compatible with any hardware platform that supports Linux, offering a versatile solution for users on their preferred hardware platform when centralized evaluation and controlling for the hardware are not primary concerns.
>
>
> > Software Environment Assumptions:
>
> Trying to maximize software flexibility is something we considered while designing the Pentathlon. Our benchmark is designed with minimal assumptions regarding the evaluated system's software environment, with the primary requirement of being able to run in a Docker container. As a quick verification, Cutlass, which the reviewer brought up, is compatible with our benchmark. Due to the limited time of the response period, the authors were not able to verify TVM at the time of this response. We plan to include this discussion in the revision.
>
> Even when the evaluated system requires compilation, its efficiency will not be negatively affected. Specifically, the Efficiency Pentathlon begins tracking efficiency metrics only once the system has completed its compilation phase and is operational.This is implemented through a 'handshake protocol': The benchmark sends a dummy input batch to the system. It will start to measure the efficiency only after it has received the system’s outputs for this dummy batch, indicating that the model has finished its preparation stage and is ready to go.
>
> > Controlled hardware vs. cloud-based platforms
>
> We fully agree with the reviewer’s great point that controlling for the hardware may not always be is not always practical or desirable in practice. This consideration is integrated into the design of our benchmark. The benchmark is developed as a standalone, open-source software, allowing users to employ it on their preferred hardware platforms, including many cloud machines that support Linux. We will emphasize this point more explicitly in the revision.

---

> > ### Author Response · Authors · 2023-11-23
> >
> > > What level of effort is required to expand Pentathlon to accommodate a new hardware platform or incorporate a new model into the benchmark?
> >
> > Expanding our centralized host machine to accommodate a new hardware platform involves physically connecting  the hardware to our host machine and making necessary software adjustments. This process can be challenging and may take 6–10 hours based on the authors’ experience. For users that want to use our benchmark on their chosen hardware,  we expect the process to involve minimal effort, typically just cloning our software and installing necessary dependencies, provided their hardware is Linux-compatible.
> >
> > To make it easier to incorporate new models, we provide code templates and detailed documentation to guide this process. Based on the experience of an author who was initially unfamiliar with the benchmark, it's estimated that an average researcher can adapt a model to our benchmark in under two hours. if the model is already compatible with Hugging Face APIs or is hosted on Hugging Face, much less effort is required.
> >
> >
> > > Beyond the BLEU score, are there additional metrics available within Pentathlon to assess model quality, such as perplexity or other relevant NLP-specific metrics?
> >
> > Efficiency Pentathlon is intentionally designed without built-in support for accuracy or quality metrics. Instead, it provides users with the models’ outputs, deferring to them conducting quality evaluations. We choose to do this for a couple of reasons.
> >
> > First, Efficiency Pentathlon is designed to support a variety of tasks, and its current version supports anything that is available on Hugging Face. Anticipating and implementing the accuracy metrics for these tasks is beyond our capabilities and the scope of this work. Second, for many tasks (e.g., MT as the reviewer mentioned), post-processing decisions can significantly affect a model's accuracy. For instance, in XX to Romanian translation, the choice to include or exclude diacritics can greatly influence the model's BLEU score. Such nuances in post-processing are critical to the overall performance evaluation. We believe that the users are better-positioned to make decisions about e.g., post processing and selecting evaluation metrics for quality. Therefore we decide to not impose specific evaluation metrics for model quality, instead leaving these decisions to the users.
> >
> > > How can you distinguish the impact of "algorithmic innovations" from other efficiency-related factors in the Pentathlon benchmark?
> >
> > The reviewer raised a great point that algorithmic innovations can be hard to distinguish from many other factors in practice. To address this, we try to control as many potential “non-algorithmic” confounders as possible, including hardware platforms, software environment, data processing pipelines, IO frameworks, etc.

---

### Official Review · Reviewer_HHde · 2023-11-01

**Soundness:** 3 good
**Presentation:** 3 good
**Contribution:** 3 good
**Rating:** 8
**Confidence:** 4

**Summary:**

This paper introduces a benchmark for evaluating model efficiency (compared with most benchmarks on performance), in particular, for LLM inference. It compares different use case scenarios for calling an LLM as well as using several metrics to evaluate the inference efficiency and environmental impacts.

**Strengths:**

* Propose several use case scenarios for model inference, like batching, streaming, offline, etc.
* Propose several metrics to measure the model inference efficiency as well as environmental impact

**Weaknesses:**

* Benchmark selection is a bit limited, only a few tasks are chosen. For instance, there is no typical (monolingual) language generation task

**Questions:**

* Figure 3 (a) and (b) are exactly the same? Is it a coincidence or mistake?

---

> ### Author Response · Authors · 2023-11-23
>
> We appreciate reviewer HHde’s recognition of our efforts to include a diverse range of efficiency scenarios and metrics.
>
> > Propose several use case scenarios for model inference, like batching, streaming, offline, etc.
>
> Our benchmark supports all tasks from Hugging Face, including many language generation tasks. Please see the general response.
>
> > Figure 3 (a) and (b) are exactly the same? Is it a coincidence or mistake?
>
> Thanks for catching this. This was due to a LaTex typo and has been fixed.

---

### Author Response · Authors · 2023-11-23
**General response**

We thank all reviewers for their thoughtful comments and constructive feedback.

Reviewers HHde, uM5U, and pV3M raised this important concern about the tasks and datasets supported in our benchmark. To clarify, Efficiency Pentathlon supports all tasks that Hugging Face does. We will highlight this in the abstract and introduction in the revision.

We chose to experiment with machine translation, classification, and math word problems in order to demonstrate Efficiency Pentathlon's capability to support diverse model architectures and settings and study their trade-off, including encoder-decoder models, decoder-only causal language models, and even non-neural approaches. Additionally, by including these varied paradigms, we can study trade-offs between techniques like CoT versus direct prompting, and zero-shot versus few-shot in-context learning.

---

### Meta-Review · Area_Chair_iYNd · 2023-12-02

**Metareview:**

This paper presents a benchmark for model efficiency.  The authors provide a controlled hardware platform for comparing different models by hosting the evaluation themselves on an in-house server.  Several types of workflows are explored, including a fully offline setting or fully online (single stream) setting.  Systems are evaluated on a suite of metrics targeting 5 aspects of efficiency: throughput, latency, memory overhead, energy consumption, and model size.

The paper conducts evaluations on MT, math reasoning (GSM8k), and classification, showcasing the kinds of insights that can be gleaned from this analysis. Models that are better on task quality metrics like BLEU score are often worse on efficiency.

The reviewers praised the overall aims of the paper: centralized benchmarking around efficiency is a valuable goal for the field.  The paper is very clearly written.

The main critiques of the paper chiefly revolve around the depth and nature of the contribution.  The authors argue in response to uM5U that this paper falls into the datasets and benchmarks track. But this is more like an *idea* of a benchmark rather than a specific benchmark; the discussion shows that different applications, different hardware, etc. are needed. For example, 5UHy brings up other hardware and software platforms that may reflect more typical use cases of systems like LLMs. uM5U argues that different tasks would be useful, and the authors respond that their benchmark can support any tasks from Hugging Face.

As a result, this paper feels more like a demo paper than an enduring research contribution.

**Justification For Why Not Higher Score:**

See review; I don't think this paper makes a research contribution suitable for ICLR.

**Justification For Why Not Lower Score:**

N/A

---

### Decision · Program_Chairs · 2024-01-16

Reject